# Double drives and private alleles for localised population genetic control

**Katie Willis** *, **Austin Burt**

Department of Life Sciences, Imperial College London, Silwood Park, Ascot, United Kingdom

* katie.willis16@imperial.ac.uk

## Abstract

Synthetic gene drive constructs could, in principle, provide the basis for highly efficient interventions to control disease vectors and other pest species. This efficiency derives in part from leveraging natural processes of dispersal and gene flow to spread the construct and its impacts from one population to another. However, sometimes (for example, with invasive species) only specific populations are in need of control, and impacts on non-target populations would be undesirable. Many gene drive designs use nucleases that recognise and cleave specific genomic sequences, and one way to restrict their spread would be to exploit sequence differences between target and non-target populations. In this paper we propose and model a series of low threshold double drive designs for population suppression, each consisting of two constructs, one imposing a reproductive load on the population and the other inserted into a differentiated locus and controlling the drive of the first. Simple deterministic, discrete-generation computer simulations are used to assess the alternative designs. We find that the simplest double drive designs are significantly more robust to preexisting cleavage resistance at the differentiated locus than single drive designs, and that more complex designs incorporating sex ratio distortion can be more efficient still, even allowing for successful control when the differentiated locus is neutral and there is up to 50% pre-existing resistance in the target population. Similar designs can also be used for population replacement, with similar benefits. A population genomic analysis of CRISPR PAM sites in island and mainland populations of the malaria mosquito *Anopheles gambiae* indicates that the differentiation needed for our methods to work can exist in nature. Double drives should be considered when efficient but localised population genetic control is needed and there is some genetic differentiation between target and non-target populations.

## Author summary

Some disease vectors, invasive species, and other pests cannot be satisfactorily controlled with existing interventions, and new methods are required. Synthetic gene drive systems that are able to spread though populations because they are inherited at a greater-than-Mendelian rate have the potential to form the basis for new, highly efficient pest control measures. The most efficient such strategies use natural gene flow to spread a construct

**Data Availability Statement:** All relevant data are within the manuscript and its Supporting Information files.

**Funding:** Supported by a grant from the Bill & Melinda Gates Foundation (Grant INV006610 "Target Malaria Phase II") and the Open

Philanthropy Project Fund, an advised fund of Silicon Valley Community Foundation (Grant O-77157) to AB. The funders had no role in study design, data collection and analysis, decision to publish, or preparation of the manuscript.

**Competing interests:** The authors have declared that no competing interests exist.

throughout a species' range, but if control is only desired in a particular location then these approaches may not be appropriate. As some of the most promising gene drive designs use nucleases to target specific DNA sequences, it ought to be possible to exploit sequence differences between target and non-target populations to restrict the spread and impact of a gene drive. In this paper we propose using two-construct "double drive" designs that exploit pre-existing sequence differences between target and non-target populations. Our approaches maintain the efficiencies associated with only small release rates being needed and can work if the differentiated locus is selectively neutral and if the differentiation is far from complete, and therefore expand the range of options to be considered in developing genetic approaches to control pest species.

## Introduction

Gene drive is a natural phenomenon in which some genes are able to increase in frequency and spread through populations by contriving to be inherited at a greater-than-Mendelian rate [1,2]. Strong drive can cause genes to increase rapidly in frequency even if they also harm the organisms carrying them, and there is currently much effort trying to develop synthetic gene drive constructs (or gene drives) to control disease-transmitting mosquitoes and other pest populations that have thus far been difficult or impossible to manage satisfactorily [3–6]. If a species is harmful and subject to control measures wherever it exists, then, in principle (i.e., in the computer), highly efficient gene drive strategies can be devised that exploit natural processes of dispersal and gene flow such that relatively small inoculative releases in a few locations can lead to substantial and widespread impacts over subsequent generations [7–9]. However, some species are pests only in a part of their range (e.g., invasive species), and other approaches are needed.

Two broad approaches have been proposed for restricting the impact of genetic control interventions to a target population. First, one can use a strategy requiring relatively large releases, which can be restricted to the target population, with any introductions into non-target populations (by dispersal, or by accidental or unauthorised releases) being too small to have a significant impact. Potentially suitable genetic constructs include those that do not drive (e.g., dominant lethals, autosomal X-shredders, or Y-linked editors; [10–12]), those that show transient drive due to a non-driving helper construct (e.g., killer-rescue systems and split drives; [13–15]) or those that drive, but only if they are above some threshold frequency (e.g., various underdominant [heterozygote inferiority] strategies, tethered drives, and split drive killer-rescue systems [16–19]). Some of these approaches are more efficient than others [12,20,21], but, by necessity, all of them require a non-trivial production and release effort.

Alternatively, if there are pre-existing sequence differences between target and non-target populations, it may be possible to exploit these differences with a sequence-specific nuclease-based gene drive that would only spread in the target population, in which case the small release rates and overall efficiency of low threshold gene drive approaches may be retained [22,23]. Sudweeks et al. [23] present useful modelling of this approach, considering the case where there is a locally fixed allele of an essential gene in the target population, while non-target populations carry a functional cleavage-resistant allele at some frequency. A single-locus gene drive that uses the homing reaction (i.e., sequence-specific cleavage followed by homologous repair [4,24]) to disrupt the locally fixed allele could be released into and eliminate the target population, but have little impact, or only a transient impact, on non-target populations. However, as emphasised by the authors, if the target population has even a small frequency of

the resistant allele, then that allele could be rapidly selected for and the intervention fail. Single locus homing drives targeting an essential gene in order to suppress a population necessarily generate strong selection pressure for resistant sequences if these can arise [24]; one approach to this problem is to target sites where functional resistance is unlikely to arise [25], but this is difficult to engineer if the target site is chosen such that a resistant allele exists at high frequency in non-target populations of the same species.

In this paper we explore alternative two-locus "double drive" low release rate strategies to restrict population control based on pre-existing sequence differences between target and non-target populations. All our designs are based on a division of labour between the two constructs, with one imposing a reproductive load by disrupting a gene needed for survival or reproduction, and therefore responsible for the desired impact (population suppression), and the other responsible for the population restriction. The first construct can then be designed to target a well-conserved essential site where functional resistance is unlikely to arise, and selection for resistant alleles at the differentiated locus will be relatively weaker because it is not directly responsible for the reproductive load. As a result, these designs are substantially less susceptible to pre-existing resistance in the target population at the differentiated locus than single drive designs, and can even work if the differentiated locus is selectively neutral. Double drives may also be useful for population replacement. Finally, analyses of published genome sequences from island and mainland populations of the malaria mosquito *Anopheles gambiae* indicates that the sort of population differentiation we model can exist in nature.

## Results

### Simple double drives for population suppression

The simplest double drive designs we consider consist of one construct (call it α) inserted into and disrupting a haplo-sufficient female-essential gene, such that homozygous females die without reproducing while heterozygous females and all males are unaffected, and a second construct (β) inserted into a sequence that is significantly more common in the target than the non-target population(s). Both constructs are able to drive by the homing reaction but α can drive only in the presence of β, while β may either drive autonomously or rely on the presence of α. With CRISPR-based designs, α would encode its cognate gRNA, β would encode the Cas9, and either construct could encode the gRNA for the second locus (Fig 1, Designs 1 and 2). We assume the α construct has been designed such that functional resistance is not possible (e.g. by targeting sequences that are essential at the nucleotide level or by using multiple gRNAs), though non-functional resistant alleles can arise by end-joining repair [25–27]. For the β construct we initially suppose its insertion site (i.e., the differentiated locus) is selectively neutral and unlinked with the α insertion site, and that differentiation is nearly complete, with the recognition sequence present at a frequency of 99% in the target population and absent in the non-target population (i.e., it is a virtually fixed private allele).

Under these conditions, a small (0.1%) release of males carrying Design 1 constructs into the target population leads to both constructs rapidly increasing in frequency and, as a result, an increasing fraction of female zygotes are homozygous for α and die without reproducing. The population size crashes to a minimum size of 3.58e-6 (relative to the pre-release equilibrium) after 25 generations (Fig 2A). Depending on the initial population size and the biology of the species (e.g., whether there are Allee effects [28] such that the population cannot persist at small sizes), this decline could be enough to eliminate the population. However, in our simple deterministic model population elimination is not possible. Instead, the population recovers due to selection of cleavage-resistant alleles at the differentiated locus (which either pre-existed or arose due to end-joining repair), leading to loss of the β construct, followed by loss

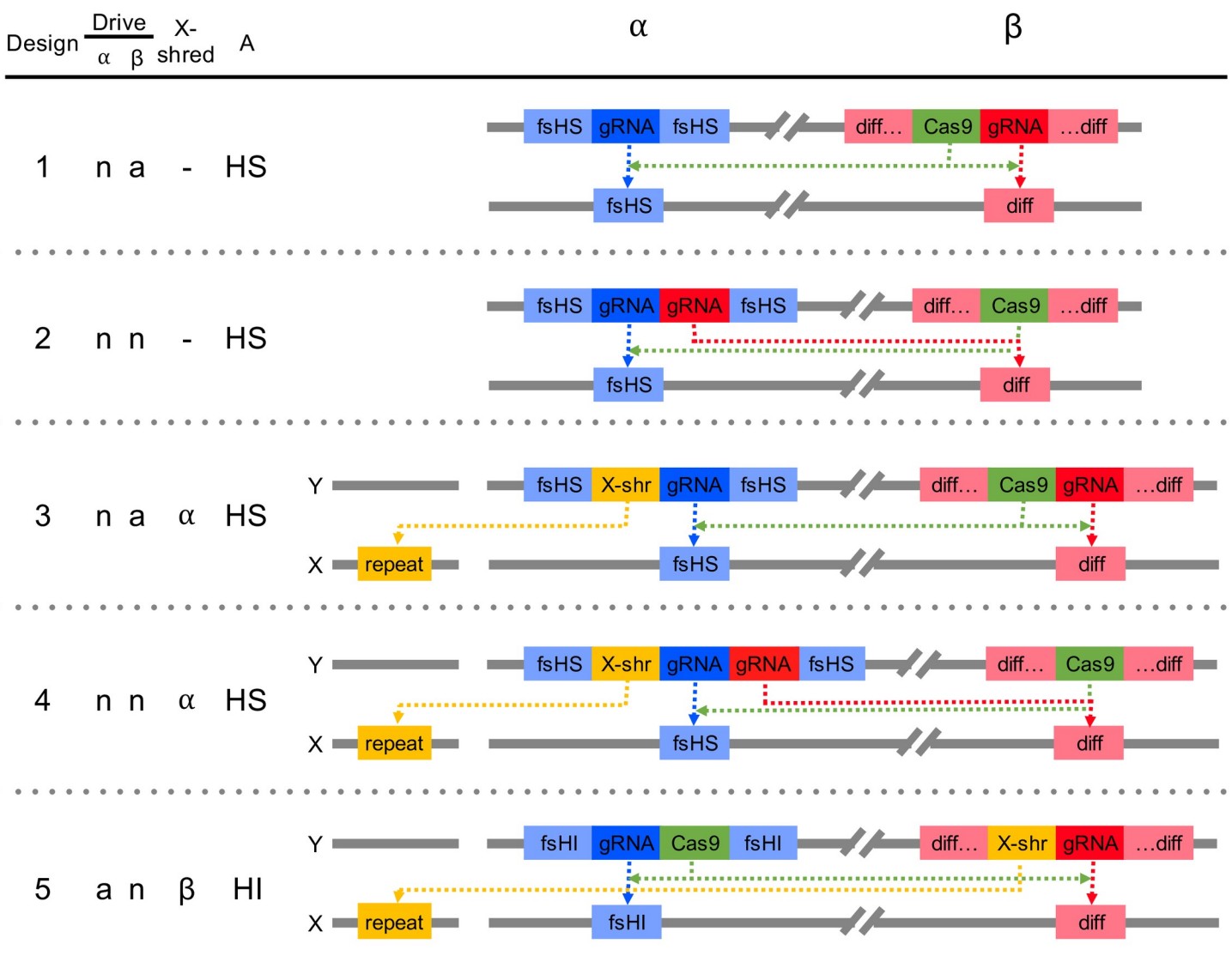

**Fig 1. Alternative double drive designs for population suppression.** Constructs α and β can drive autonomously (a) or non-autonomously (n); one or the other may encode an X-shredder; and the A target locus can be a gene that is haplo-sufficient (HS) or haplo-insufficient (HI) for female viability or fertility. fsHS—female-specific haplo-sufficient locus; fsHI—female-specific haplo-insufficient locus; diff—differentiated sequence; X-shr—X-shredder targeting an X-linked repeat.

of α, allowing the wild-type allele and population fertility to be restored. By contrast, the same releases into the non-target population have minimal effect: β cannot increase in frequency (because its target site is absent), and therefore α remains rare, and population size is little affected (Fig 2B).

Because the spread of construct α in the target population depends on β, and therefore will be affected by the association between them, it might be expected that close linkage between the two constructs may increase construct spread and the extent of population suppression. Close linkage has been observed to affect the dynamics of other two-locus drive systems [15]. Furthermore, because population recovery (if it occurs) will be due to the evolution of resistance at the differentiated locus, additional improvements might be expected by using an essential gene as the differentiated locus while designing β to have minimal fitness effects (e.g.,

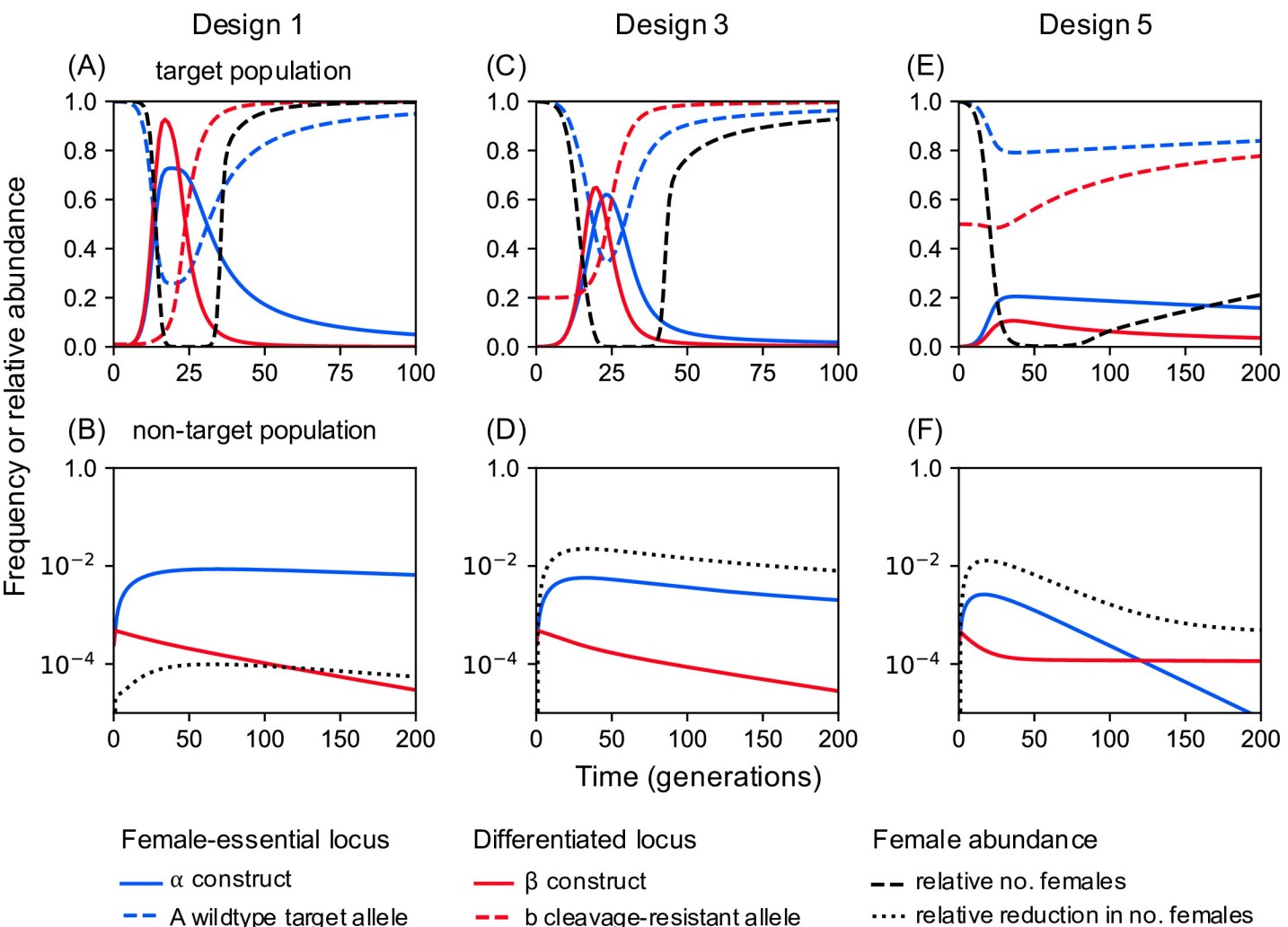

**Fig 2. Performance of double drives for population suppression.** **(A, B)** Timecourse for Design 1 in target and non-target populations, assuming 1% and 100% pre-existing resistance at the differentiated locus, respectively. In the target population the α and β constructs increase in frequency together (blue and red solid lines), causing the number of females to decline. If the population is not eliminated, then eventually the resistant b allele replaces β, followed by the wild-type A allele replacing α, allowing the population to recover. In the non-target population both constructs remain rare and the reduction in female numbers remains small. **(C, D)** Timecourse for Design 3 assuming 20% and 100% pre-existing resistance in the target and non-target populations. **(E, F)** Timecourse for Design 5 assuming 50% and 100% pre-existing resistance in the target and non-target populations.

by containing a recoded, cleavage-resistant version of the essential gene [14,26,29], or by being inserted in an artificial intron [30]). End-joining repair will then tend to produce non-functional resistance alleles, increasing the load, and functional resistance at the differentiated locus will be slower to evolve, relying instead on pre-existing resistant alleles. Moreover, this effect may be stronger if the essential gene is haplo-insufficient than if it is haplo-sufficient, as found with some other gene drive designs [31]. Both these expectations about linkage and using an essential differentiated gene are met individually, and, in combination, can reduce the minimum population size achieved by many orders of magnitude (Fig 3; see also S1 Fig for the separate effect of each modification). If it is not possible to have close linkage, then the maximum level of suppression can also be increased by releasing the two constructs in different males rather than in the same males, which allows β to escape some of the fitness costs of α and get to a higher frequency than it otherwise would, though at the cost of the impact being

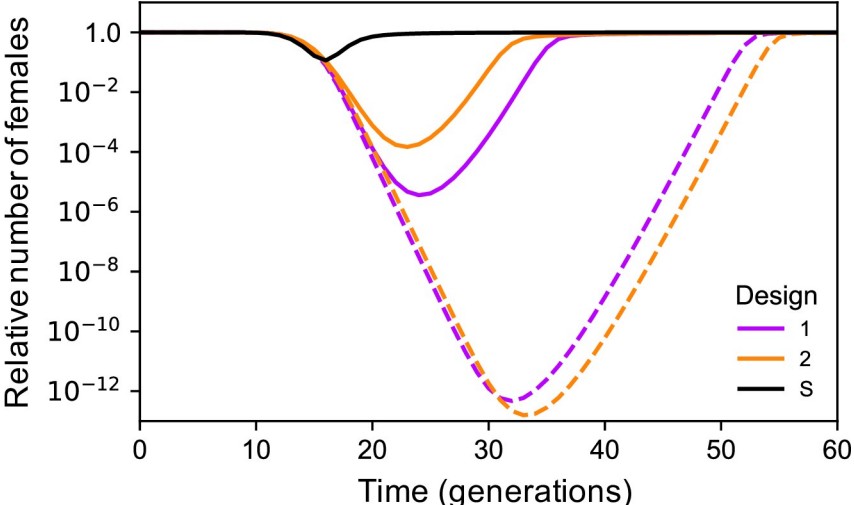

**Fig 3. Timecourse for the relative number of females over time for Designs 1 and 2.** Solid lines are for β in a neutral locus unlinked to the α construct, and dashed lines for β as a neutral insertion in a haplo-insufficient essential gene closely linked (r = 0.01) to the α construct. In all cases there is 1% pre-existing resistance at the differentiated locus. Also shown for comparison are results for a single construct drive targeting a haplo-sufficient female-specific viability gene, assuming 1% pre-existing functional resistance and that end-joining repair produces only nonfunctional resistant alleles (S).

delayed, and separate releases perform worse than combined releases when linkage is tight (S2 Fig).

Design 2, which has the same components as Design 1, but arranged differently such that homing of the β construct only occurs in the presence of α, has dynamics qualitatively similar to Design 1, but quantitatively different (S3 Fig). Interestingly, if the two constructs are unlinked then the extent of suppression is less than with Design 1, but if they are closely linked then the suppression can be greater (Figs 3 and S1). For comparison we also model a single drive homing into a differentiated female-essential gene with 1% pre-existing resistance. The maximum extent of suppression is much less than with any of the double drives considered, because selection for resistance is much stronger, being directly at the fitness-determining locus (Fig 3).

## Coping with higher frequencies of pre-existing resistance

Though these simple double drive designs work well with 1% pre-existing target site resistance at the differentiated locus, performance declines rapidly after that. For example, if there is 10% pre-existing resistance, then even the best of these designs (Design 2 with close linkage and the differentiated locus being haplo-insufficient) only suppresses the target population to a minimum of 2.38e-4 (Fig 4). In some situations the target population may not have a private allele with frequency over 90% and alternative approaches would need to be considered. One possibility is to increase the load imposed on the population by the α construct by adding to it an X-shredder locus that destroys the X-chromosome during spermatogenesis such that α now produces a male-biased sex ratio as well as killing homozygous females (Fig 1, Designs 3 and 4). Since population productivity in many species depends on the number of females, population size may thereby be further reduced. A single drive based on these components has previously been constructed in *Anopheles gambiae* by Simoni et al. [32]. Our modelling indicates that adding an X-shredder to a double drive gives a quantitative improvement in the dynamics, and even pre-existing resistance frequencies of 20% are compatible with good control, while still having minimal effect on non-target populations (Fig 2C and 2D).

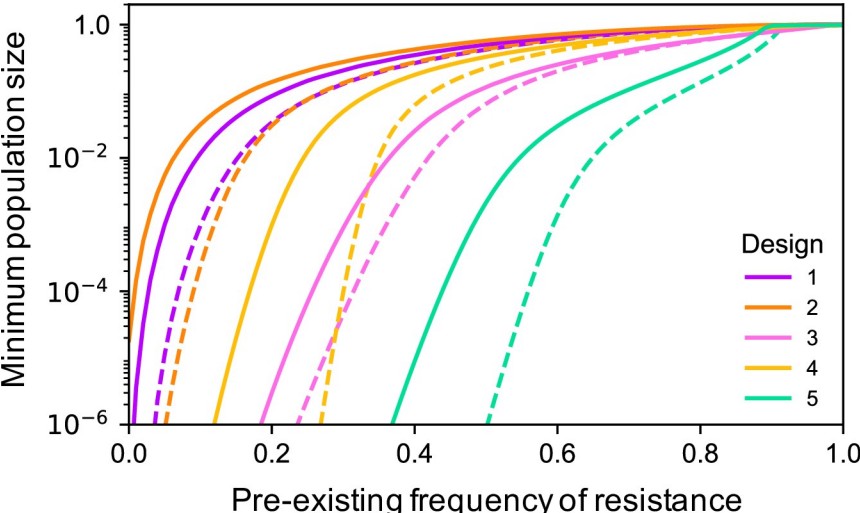

**Fig 4. Minimum population size for each of the 5 designs as a function of the pre-existing frequency of resistance at the differentiated locus.** Solid lines are for the baseline case (r = 0.5, β in a neutral locus), while dashed lines are for the improved case (r = 0.01, β as a neutral insertion in a haplo-insufficient essential gene).

Even more robust control can be obtained by adding the X-shredder to the β construct and having the α construct drive autonomously in males and cause dominant sterility or lethality in females (e.g., target a female-specific haplo-insufficient locus; Fig 1, Design 5). As the Cas9 is encoded by α, homing of β also occurs only in males. The dynamics in this case are somewhat different from the others: the X-shredder does not function to directly increase the load, but instead it allows the α construct to spread in the population, because it will end up more often in males (where it homes), and less often in females (where it is a dead end). The male bias also protects the β construct from the female lethality produced by the α construct, and so selection against β is much weaker than in the previous designs, and resistance evolves more slowly (compare the rate of spread of the resistant b allele in Fig 2E to that in Fig 2A and 2C). As a result the design is able to perform well even with pre-existing resistance of up to 50%, but still not spread in the non-target population (Fig 2E and 2F). Moreover, if the population is not eliminated, it can nevertheless be suppressed for many generations. For example, with 50% pre-existing resistance the minimum population size reached is 2.15e-3, and the population remains below 5% of its pre-intervention size for 63 generations; with close linkage (r = 0.01), then the corresponding values are 8.79e-7 and 147 generations. A comparison of the maximum extent of suppression as a function of the pre-existing resistance frequency for the different designs is shown in Fig 4 (see also S4 Fig). Note that none of the modifications considered (linkage, use of an essential differentiated gene, or separate releases) has a qualitative effect on dynamics in the non-target population, as β is still unable to increase in frequency, and impacts on population size remain small (S5 Fig).

## Evolutionary stability and impact of fitness costs

We now explore the consequences of relaxing two assumptions that have been implicit thus far in our modelling. First, we have assumed that our various constructs remain intact after release. In fact, mutations that destroy the function of one component or another will be expected to arise as the constructs spread through a population, particularly as homing may be associated with a higher mutation rate than normal DNA replication [33–35]. For components that contribute directly to their construct's spread, one would expect that loss-of-function

mutations would remain rare in the population and have little effect, whereas for other components (e.g., the X-shredder, especially in Designs 3 and 4), such mutations may be actively selected for. To investigate we allowed homing-associated loss-of-function mutations to occur in each component of each construct. Mutation rates of 10e-3 have a small but significant impact on the performance of the three designs with an X-shredder, due to the accumulation of mutant constructs missing that component, while mutation rates of 10e-4 have negligible impact for all designs (Figs 5A, S6 and S7).

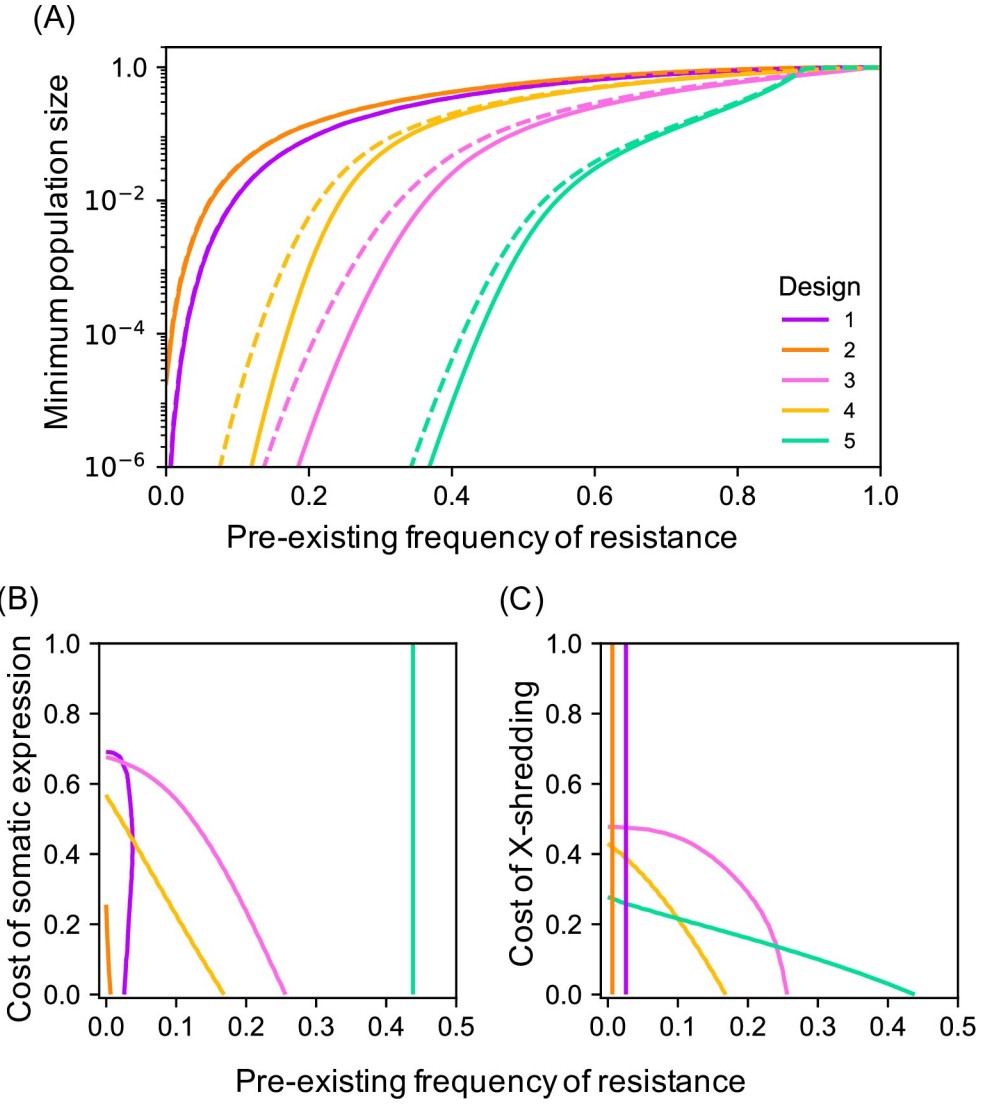

**Fig 5. The impact of evolutionary stability and added fitness costs on the performance of each of the 5 designs for population suppression. (A)** The effect of loss-of-function mutations on the minimum population size (number of adult females) reached. Solid lines are for the baseline case of no mutations, and dashed lines are with each component of each construct having a mutation rate of 10e-3 per homing event. Note that the effect is only visible for designs with an X-shredder, and if the mutation rate was 10e-4, the results for all designs would be virtually indistinguishable from the solid lines. **(B, C)** Contour plots showing combinations of fitness costs and pre-existing frequency of resistance giving a minimum population size of 10e-4 for different double drive designs. For **(B)** the costs are reductions in female fitness due to somatic expression of the nuclease targeting the A locus, and for **(C)** the costs are reductions in male fitness due to the X-shredder. Vertical lines indicate the cost is irrelevant, either because heterozygous females in any case have fitness 0 (Design 5 in **(B)**), or because the designs do not include an X-shredder (Designs 1 and 2 in **(C)**).

Second, we have assumed thus far that the genetic constructs have little unintended impact on survival or reproduction. Experiments with *An. gambiae* have revealed at least two unintended fitness costs can occur, a reduced fitness of homing heterozygous females due to somatic expression of the nuclease [25,32], and reduced fitness of males expressing an X-shredder, possibly due to paternal deposition of the nuclease and/or reduced sperm production [36]. The first of these costs is not relevant to Design 5 (because heterozygous females die anyway), and the second is not relevant to Designs 1 and 2 (because they do not use an X-shredder), but in other contexts, as expected, these costs reduce performance, requiring a lower frequency of pre-existing resistance in order to achieve a particular level of suppression (Fig 5B and 5C).

## Population replacement

Gene drive can be used not only for population suppression but also to introduce a new desirable 'cargo' gene into a target population for population replacement or modification–for example, a gene reducing a mosquito's ability to transmit a pathogen [37,38]. In double drive designs for population replacement the α construct would carry the cargo and homing by α would require β, while that by β could be either autonomous or depend on α (analogous to Designs 1 and 2 for population suppression; Fig 6A). Both α and β could be inserted into neutral sites, or into essential genes in such a way as to minimise fitness effects. We have modelled these approaches assuming, for purposes of illustration, the cargo imposes a dominant 20% fitness cost on females, and find that, again, such double drives can spread rapidly through target populations even when there is significant pre-existing resistance, and would not spread in non-target populations fixed for the resistant allele (Fig 6B and 6C). Unless there is virtually no pre-existing resistance at the differentiated locus, double drives can keep the frequency of the cargo gene above 95% much longer than a single drive construct targeting a differentiated locus, either neutral or essential (Fig 6D). In the single locus case selection rapidly increases the frequency of a pre-existing functional resistant allele because there is both significant variation and significant fitness differences (arising from the cost of the cargo gene) at the same locus. By contrast, in the double drive case there is one locus at which there are fitness difference (due to presence/absence of the cargo) but very little variation (initially none, and arising only after release due to rare end-joining and loss-of-cargo events), and another (differentiated) locus at which there is pre-existing variation but much smaller fitness differences (arising only due to the statistical correlation between alleles at the two loci). Finally, as with double drives for population suppression, the protection provided by a double drive for replacement can be extended even further if the two loci are tightly linked and α is inserted in an essential gene, with end-joining repair producing nonfunctional alleles (Fig 6D). In the latter case the only source of functional cleavage-resistant cargo-less alleles are the constructs that lose their cargo during homing, and, since this is assumed to occur at a much lower rate than end-joining repair, more generations are needed for such alleles to become common and the duration of protection is extended.

## PAM site analysis in An. gambiae

To explore whether the type of population differentiation assumed in our modelling can exist in nature, we analysed published genome sequence data on *An. gambiae* mosquitoes from the Ag1000G project [39]. The Ag1000G dataset includes sequences from 16 mainland African populations and from populations on Mayotte and Bioko, two islands 500km off the east and 30km off the west coast of Africa, respectively. Note that in presenting this analysis we are not advocating the use of double drives on these islands, and merely wish to investigate whether

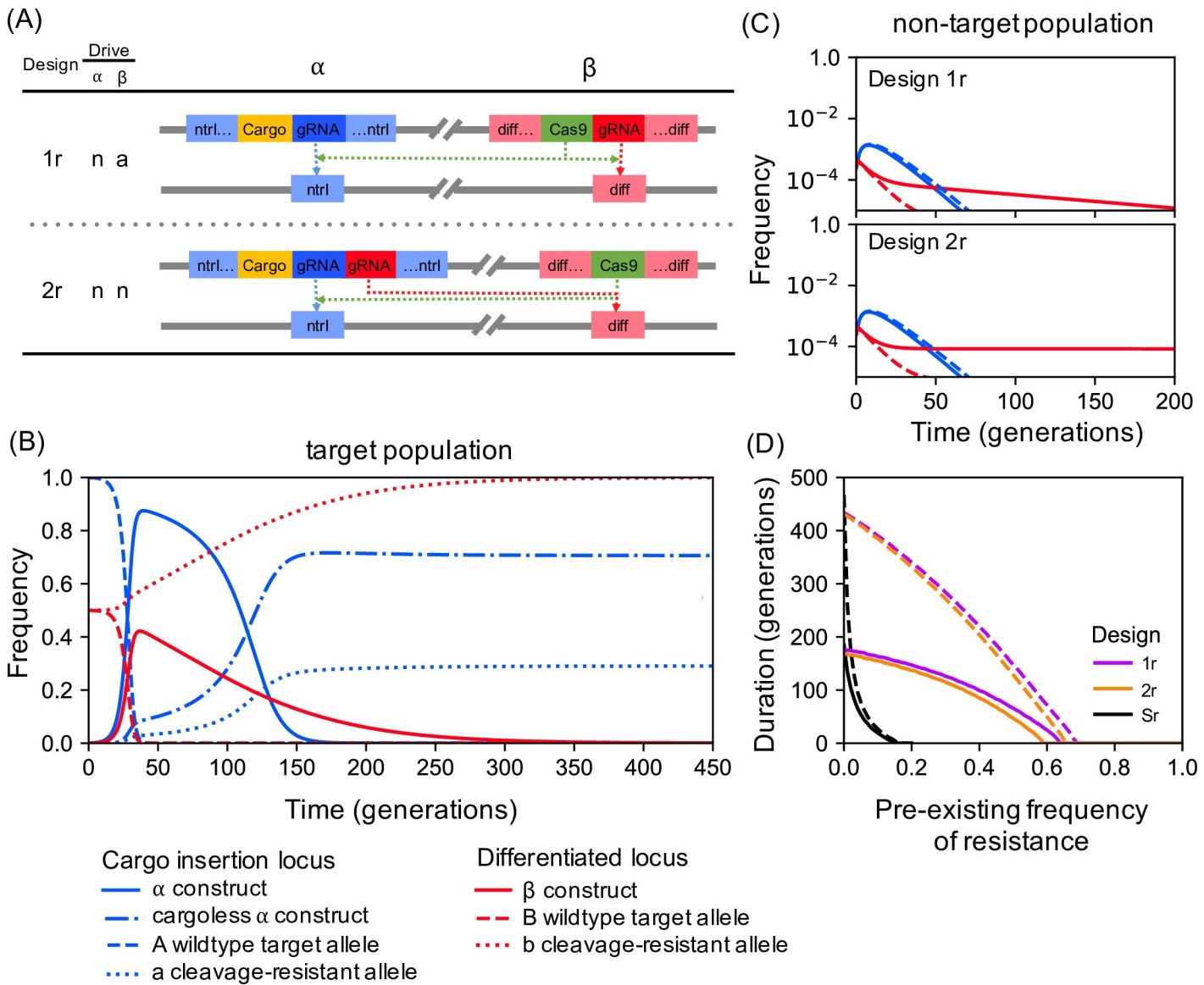

**Fig 6. Double drives for population replacement.** **(A)** Alternative double drive designs. **(B)** Timecourse of allele frequencies for Design 2r in a target population assuming 50% pre-existing resistance; "cargoless α construct" refers to α constructs that have lost their cargo gene. The dynamics for Design 1r are qualitatively similar. **(C)** Allele frequencies for Designs 1r and 2r in a non-target population with 100% pre-existing resistance, assuming insertion of both α and β into neutral unlinked loci (solid lines); or α as a neutral insertion into a haplo-insufficient essential gene closely linked to β (r = 0.01) (dashed lines). **(D)** Duration of at least 95% of adult females carrying a cargo gene (whether the rest of the construct is functional or not) as a function of the pre-existing frequency of resistance at the B locus for double drives 1r and 2r, where solid and dashed lines are as in **(C)**. Also shown for comparison are results for a single drive (S) carrying the cargo at a neutral A locus (solid line) or a haplo-insufficient essential gene assuming end-joining repair produces only nonfunctional resistant alleles (dashed lines). Note that results for insertion of β into a haplo-insufficient essential gene would be virtually indistinguishable from the solid lines **(C, D)**. All plots assume 20% fitness cost of the cargo on females and a homing-associated loss-of-function mutation rate of 10e-3.

the requisite differentiation can be found on island populations. For our analysis we focussed on potential PAM sequences (NGG or CCN), on the logic that a construct would be unlikely to mutate to recognise a new PAM, whereas this could occur for a protospacer. The entire dataset includes 57 million polymorphic sites, which we screened for PAM sites present in the island population and at a frequency <10%, <5%, or absent from all other populations. In Mayotte, for PAM sequences that were completely private to the island (i.e., not found in any

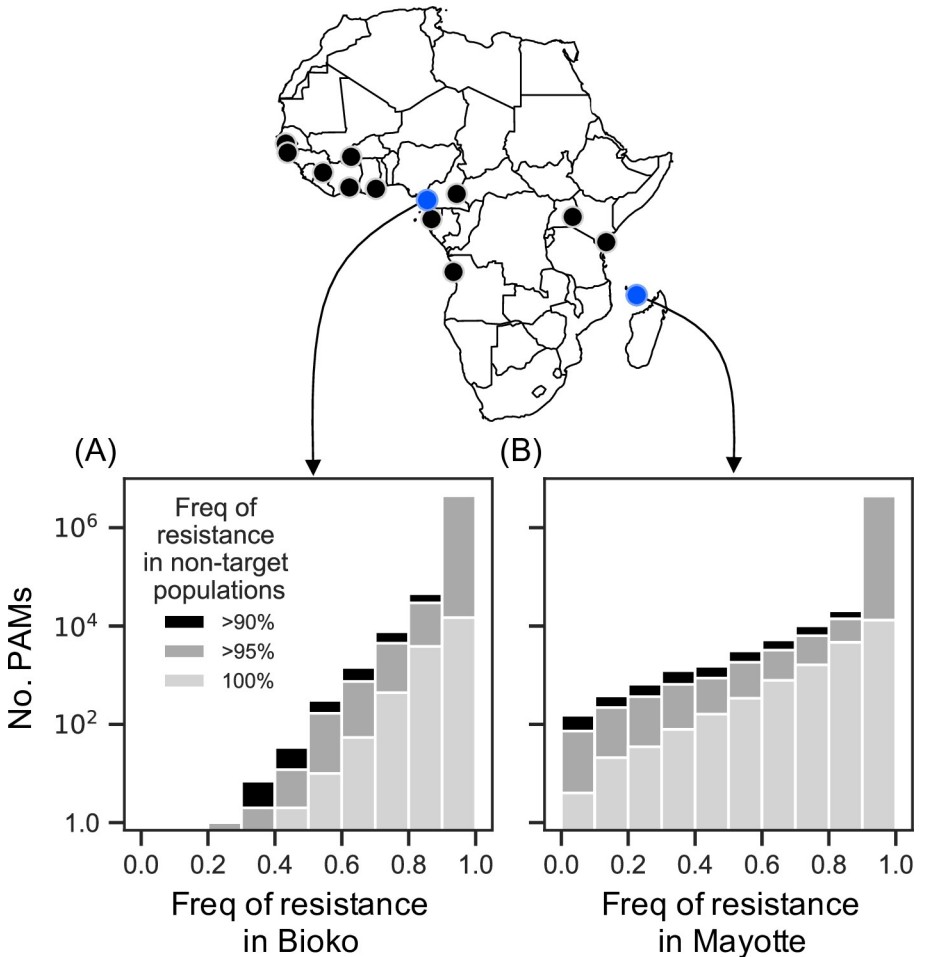

**Fig 7. Frequency of PAM sites in island populations of An. gambiae.** Numbers of PAM sites (NGG or CCN) with varying frequencies of resistance in samples of *An. gambiae* from two oceanic islands (blue map circles): **(A)** Bioko island (n = 18 sequences, from 9 individuals), and **(B)** Mayotte island (n = 48 sequences, from 24 individuals), where the PAM site frequency in each non-target population (black map circles) is <10%, <5% or 0% (i.e., target site resistance is >90%, >95% or 100%). GG or CC dinucleotides which varied by at least one base were considered to be resistant.

other population), only 1 of them had no pre-existing resistance (i.e., was found in all 48 sequences from the island), whereas 25 had pre-existing resistance less than 20%, and 353 had pre-existing resistance less than 50%. PAM sequences with small but nonzero frequencies on the mainland were even more abundant (Fig 7). Bioko island is not as differentiated as Mayotte from the mainland populations, and the sample size is smaller (18 sequences), but still there are some potential candidate sites.

## Discussion

Given that some of the most promising gene drive approaches for population control use (CRISPR-based) sequence-specific nucleases, an obvious way to limit their spread and impact is to exploit sequence differences between target and non-target populations. In this paper we have proposed using a double drive design, here defined as one that uses two constructs, inserted at different locations in the genome, both of which can increase in frequency, at least initially, and which interact such that the transmission of at least one of them depends on the

other. Previously published examples that fit this definition include those for 2-locus under-dominance [16,19,40,41], and Medusa [42], tethered [17], integral [30], and transcomplementing [43] gene drives. As with single-construct gene drives, these various proposed designs differ in purpose (suppression vs. modification), release rate needed to initiate spread (low vs high threshold), and the molecular basis for the superMendelian inheritance (homing, toxin-antidote interactions, or a combination of the two), and the suggested rationales for adopting these designs over single drives include allowing more localised population control and a more modular product development pipeline. The requirement that both constructs can increase in frequency over time excludes split drives [14,44–46] and killer-rescue systems [13,47], in which only one of the two components increases in frequency. In our proposed designs there is a division of labour between the two constructs, with one responsible for the desired impact (suppression or replacement) and the other for the population restriction, such that together they act as a double drive in the target population and as a split drive in non-target populations.

Note that if there are multiple populations of the same species requiring control, each with a different private allele, the same α construct could be used in each case, with only a change in the insertion site of the β construct and the corresponding gRNA. This flexibility may be particularly useful when the α construct requires significant optimisation [30]. Moreover, the same strategies may also be useful for controlled suppression of a target population even when there is no concern about non-target populations (and therefore no need to target a private allele): by appropriate choice of construct components and insertion sites a form of "planned obsolescence" could be achieved, with a wider and more predictable range of suppression profiles (e.g., depth and duration of suppression) possible than with conventional single drives [see also [48]].

For designs that involve interacting insertions at two or more loci, their population genetic dynamics and impact will usually depend on the statistical correlation between the constructs, and therefore also on the degree of linkage between them [12]. As previously demonstrated for split cleave-and-rescue designs by Oberhofer et al. [15], the degree of linkage between constructs can therefore be used as a tunable parameter to control the dynamics. They found that the expected impacts of a release were stronger and longer-lasting with closely linked constructs than with distantly linked ones, and we found much the same with our double drive designs, though there are some differences in detail between the systems. In particular, with split cleave-and-rescue designs, which do not rely on homing, if there is complete linkage (i.e., no meiotic recombination) then the system behaves the same as a single locus construct, whereas that is not the case for our homing-based double drives, where constructs can be separated if one homes and the other does not, even if there is never any meiotic crossing over between them. Thus, in our model, setting r = 0 (while allowing separate homing events) does not reproduce the single drive dynamics. If the insertions are physically very close to each other, then there may be some direct mechanistic interaction between them (e.g., binding of one nuclease complex preventing the other from binding, or resection of DNA during the repair process leading to co-homing of the two insertions), or simultaneous cleavage may lead to a large deletion. We have assumed that our constructs are far enough apart as not to interact in this way (e.g., r = 0.01 corresponds on average to between 600kb and 1Mb in *An. gambiae* [49]).

We have considered a range of double drive designs of increasing robustness, as judged by their ability to cope with an increasing frequency of pre-existing resistance at the differentiated locus. The simplest designs do not have any component beyond those needed for any CRISPR-based construct, and so should be widely applicable [43]. More powerful constructs can be made by adding an X-shredding sex ratio distorter to the load-inducing construct;

these have been most effectively demonstrated in *An. gambiae* mosquitoes [11,50], but may also work more broadly [51]. Note that the optimal timing of homing and X-shredding during gametogenesis may be different, requiring different and compatible control sequences, which will need to be taken into account in construct design [50,52]. The ability to control gRNA expression in a tissue-specific manner would be helpful in this regard. In other species there are other ways to distort the sex ratio [53–55], and it would be interesting to model whether these alternatives would be expected to have the same impact as an X-shredder in the context of a double drive. Potentially an even simpler way to increase the load may be to include additional gRNAs in the α construct that cleave and knock out the function of other female fertility genes elsewhere in the genome [31,56]. The effect of such gRNAs would depend on the heterozygous and homozygous fitness effects of the mutations caused and, again, on the degree of linkage with the target site, and further modelling would be needed to investigate whether such a strategy is worthwhile. The most powerful design we considered targets a female-specific haplo-insufficient gene, or otherwise causes dominant female sterility or lethality. Such genes are not common, but there are some possible candidates [57–60], and our modelling motivates the search for others. Finally, performance (in terms of being able to cope with ever higher frequencies of pre-existing resistance) could presumably also be improved by using a third construct, to construct a triple drive, though modelling would be required to explore the implications of the many different configurations this extension would allow.

The proposed strategy requires that there be a differentiated locus between target and non-target populations. It need not be an essential gene, and could even be selectively neutral. Our focus has been on using so-called private alleles–sequences that are present (but not necessarily fixed) in the target population, and absent (or of negligible frequency) in non-target populations. Our analysis of PAM sites in *An. gambiae* indicates that appropriately differentiated sites may exist in island populations of this species, though our analysis must be considered preliminary: the dataset does not include mainland sites in closest proximity to the island populations, where differentiation may be lower, and we have not considered potential polymorphism in the protospacer sequence (which, if present, may require the use of multiple gRNAs). We have focussed on nucleotide variation at PAM sites on the assumption that a construct is unlikely to mutate to recognise a new PAM; structural variation in the protospacer region may also be an appropriate basis for geographically restricting double drive spread. We have also not attempted to determine whether the observed differentiation is due solely to mutation and drift, or if selection may be involved as well.

Note that the single drives modelled by Sudweeks et al. [23] require the opposite type of differentiation: sequences that are fixed in the target population, even if not private (i.e., even if found at appreciable frequencies in the non-target population [61]). In this latter scenario the challenge is not so much to have an impact on the target population as to not have an impact on the non-target population. What constitutes "acceptable non-impact" may differ widely from one use case to another and must be assessed on a case-by-case basis: in some circumstances spread of the construct and a transient decline in population size followed by recovery may be acceptable, whereas in others any significant spread of the construct may be unacceptable, regardless of impact on population size. Designs with non-autonomous homing of the β construct (Designs 2, 4, and 5) should be less likely to increase in frequency in the non-target population, and may therefore be preferable. We have focused in this paper on differentiated loci on autosomes, but note that for Design 5 the X-shredder is required for the spread of the α construct and, in principle, one could achieve population-restricted spread if the shredder targeted a population-specific sequence on the X chromosome (rather than inserting it into a population-specific autosomal sequence). In many species the X chromosome shows greater population differentiation than autosomes [62], so this alternative may be useful. Finally, if

there are no private alleles in the target population, it may be worthwhile considering a two-step approach of first introducing a private allele into a population and then using that allele to control the population [22]. The ability of double drives to exploit private alleles that are selectively neutral and that have a frequency of only 50% (suppression) or 20% (modification) potentially makes this approach more feasible than would otherwise be the case.

In this paper we have used a simple high-level modelling framework in which the generations are discrete, the population is well mixed, and dynamics are deterministic. This framework is appropriate for strategic models aiming to identify candidate approaches that are worthy of further investigation. For any specific use case the appropriate tactical models would need to be developed that incorporate more biological detail, including spatial and stochastic effects. In spatially distributed populations with local mating the statistical association between alleles at the two loci may differ from that in our well mixed model, and the quantitative dynamics thereby affected. Issues of evolutionary stability and the breakdown of constructs can also be more important in such models, as previously demonstrated for single drive homing constructs for population replacement [63]. Such extensions will be particularly important when the goal is to eliminate the target population, which is not possible in our deterministic models. Instead, we have reported the minimum relative population size achieved, which is expected to be related to the size of a population that could be eliminated, but determining the precise connection will require bespoke modelling tailored to a specific situation. Further extensions would be needed to allow for on-going movement between target and non-target populations–if there is on-going immigration into the target population, and this cannot be stopped, then it may not be possible to eliminate the target population with a single release of a double drive. Nevertheless, such a release may be sufficient to suppress the population to such an extent that it can be controlled by other means, including recurrent releases of the same constructs. If one is able to achieve an initial release rate of 1% into a target population, and that suppresses the population by a factor of 1000, then the same releases going forwards will constitute a 10-fold inundation, and self-limiting genetic approaches may be sufficient.

## Methods

The basic deterministic modelling structure follows that of Burt & Deredec [12]. In brief, populations have discrete generations, mating is random, there are two life stages (juveniles and adults), and juvenile survival is density dependent according to the Beverton-Holt model with an intrinsic rate of increase ($R_m$) equal to 6 [56]. Genetic parameter values (rates of DNA cleavage, rates of alternative repair pathways, and the sex ratio produced by X-shredding) are as estimated from *An. gambiae* (S1 Table) [11,25,52]. Constructs may be inserted into a haplo-sufficient or haplo-insufficient female-essential gene (in which case gene function is disrupted), a selectively neutral sequence (in which case the insertion is also selectively neutral), or a haplo-sufficient or haplo-insufficient gene required for male and female viability (in which case the insertion is again selectively neutral, because it contains a re-coded version of the target gene [14,26,29], or is inserted in an artificial intron [30]). For constructs inserted into an essential gene we assume end-joining repair produces non-functional cleavage-resistant alleles [25,64], while for constructs inserted into selectively neutral sites the products of end-joining repair are also neutral. In all models we assume individuals with an intact CRISPR system suffer a 1% fitness cost for every different gRNA they carry as a cost of off-target cleavage, and for population replacement we assume the cargo gene imposes a 20% fitness cost on females. Both these costs are assumed to be dominant (i.e., not dosage-dependent). For simplicity, we assume all fitness costs affect survival after density dependent juvenile mortality and before censusing (e.g., as if pupae die). All results are for populations censused at the adult

stage. Releases are of heterozygous adult males at 0.1% of the pre-release number of males, and if the two constructs are linked then they are assumed to be in cis; for constructs released in separate males we assume release rates of 0.05% of each. Additional details and a list of parameters and their baseline values is given in S1 Text, S1 and S2 Tables. Code for implementation of the simulations is available on GitHub (https://github.com/KatieWillis/DoubleDriveSimulator).

For the PAM site analysis we screened the Ag1000G phase II SNP data for PAM sites (GG or CC dinucleotides) showing variation between samples at one or both nucleotides. PAM site frequencies were calculated per sampling location and filtered for those present in the island population and at <10%, 5%, or absent from all other populations, excluding those containing >5% missing data in at least one sampled population. Further details are given in the S1 Text. The map in Fig 7 was produced using the cartopy python package [65].

## Supporting information

**S1 Text. Supplemental methods.**
(PDF)

**S1 Table. Model parameters and baseline values.**
(PDF)

**S2 Table. Host gene disruption fitness costs.**
(PDF)

**S1 Fig. Timecourse for the relative number of females over time for Designs 1 and 2.** Solid lines are for where α and β are unlinked and dashed lines for where they are linked (r = 0.01). Shown are the cases where β is inserted as a neutral insertion into **(A)** a neutral locus, **(B)** an essential haplo-sufficient gene or **(C)** an essential haplo-insufficient gene. Shown for comparison is a time course for a single drive targeting a haplo-sufficient female-specific viability gene (S).
(TIF)

**S2 Fig. Effect of releasing constructs in the same or different males.** Comparison of dynamics for Design 1 when the constructs are unlinked and are released in the same **(A)** or in different **(B)** males. If the constructs are released in separate males the initial correlation between α and β (black dotted line) is negative, allowing β (solid red line) to increase to a higher frequency than if released in the same males as α where it experiences higher fitness costs. Consequently α (solid blue line) is retained at high frequency in the population for longer resulting in a greater reduction in relative number of females, though there is a longer delay between release and impact. **(C-G)** Timecourse for the relative number of females over time for Designs 1–5 where constructs are unlinked and released in the same males (solid lines), linked and released in the same males (dotted lined), unlinked and released in different males (dashed lines) or linked and released in different males (dot-dashed). Pre-existing resistance is assumed to be 1% **(C, D)**, 20% **(E, F)** and 50% **(G)**. For Designs 2, 4 and 5 **(D, F, G)** separate releases only delay the impact because β cannot increase in frequency autonomously, whereas for Designs 1 and 3 separate releases can give a larger (though still delayed) impact when constructs are unlinked, but not when they are closely linked. Shown for comparison is a time course for a single drive targeting a female-specific viability gene with the same level of pre-existing resistance (1%, 20%, or 50%; black solid lines).
(TIF)

**S3 Fig. Example time courses for double drives for population suppression.** Design 1 **(A, B)** and 2 **(B, C)** assuming 1% and 100% pre-existing resistance in target and non-target

populations. Design 3 (**E, F**) and 4 (**G, H**) assuming 20% and 100% pre-existing resistance in target and non-target populations. Design 5 (**I, J**) assuming 50% and 100% pre-existing resistance in target and non-target populations respectively. Plots for Designs 1, 3, and 5 are the same as in the main text, and presented here to facilitate comparisons.
(TIF)

**S4 Fig. The impact of alternative designs and their variants on population suppression as a function of the pre-existing frequency of resistance.** Solid lines are for baseline conditions (α and β are released in the same males, β is in a neutral locus, and loci are unlinked), and are the same in each panel. Dashed lines are for variants where (**A**) β is inserted as a neutral insertion into an essential haplo-insufficient gene, (**B**) β is inserted as a neutral insertion into an essential haplo-sufficient gene, (**C**) loci are linked (r = 0.01), and (**D**) α and β are released in separate males, holding all other properties at baseline.
(TIF)

**S5 Fig. Timecourse of allele frequencies and population suppression (1-relative number of females) for Designs 1–5 in non-target populations where the resistant allele is present at 100%.** (**A-E**) α and β are unlinked (solid lines) or linked (dashed lines). (**F-J**) β is inserted into a neutral site (solid lines). Note that the effect of inserting β as a neutral insertion into a haplo-sufficient or haplo-insufficient essential gene would be virtually indistinguishable from the solid lines. (**K-O**) α and β are released in the same males (solid lines) or different males (dashed lines).
(TIF)

**S6 Fig. Timecourses for Designs 3, 4 and 5 assuming pre-existing frequency of resistance of 1%, where homing-associated loss-of-function mutations occur for each component of each construct with probability 10e-3.** For each design, the intact constructs (α, blue solid lines and β, red solid lines) increase in frequency together, causing the relative number of females (black dashed lines) to decline. For designs 3 and 4, loss-of-function mutations at the X-shredder (α-XS, blue dashed-dotted lines) are selected for, replacing α (**A, C**). Since α-XS is identical to α in designs 1 and 2, the construct continues to reduce the relative number of females. If the population is not eliminated, β is eventually replaced by the resistant b allele and α-XS is replaced by the wild-type A allele, allowing the population to recover. For design 5, loss-of-function mutations at the X-shredder (β-XS, red dashed-dotted lines) are also selected for, but increase in frequency more slowly than the α-XS allele in Designs 3 and 4, resulting in the intact β construct persisting for longer. For all designs, loss-of-function mutations at each of the other components (Cas-9, gRNA$_A$ and gRNA$_B$) remain at low frequency, having negligible impact on the efficacy of the designs (**B, D, F**). Note that these results are not directly comparable to Figs 2 and S3 due to differing pre-existing resistance frequencies.
(TIF)

**S7 Fig. Loss-of-function mutation rates of 10e-4 have minimal impact on the extent of population suppression.** Solid lines are for baseline conditions where constructs remain intact after release, while dashed lines are for homing-associated loss-of-function mutations occurring at each component of each construct with probability 10e-4.
(TIF)

## Acknowledgments

We thank Alistair Miles, Nick Harding and Christopher Clarkson for advice on analysing the Ag1000G data, John Connolly, Silke Fuchs and John Mumford for useful comments on a previous draft and Vassiliki Koufopanou for valuable discussion.

## Author Contributions

**Conceptualization:** Austin Burt.

**Data curation:** Katie Willis.

**Funding acquisition:** Austin Burt.

**Investigation:** Austin Burt.

**Methodology:** Katie Willis, Austin Burt.

**Software:** Katie Willis.

**Supervision:** Austin Burt.

**Validation:** Katie Willis, Austin Burt.

**Visualization:** Katie Willis.

**Writing – original draft:** Austin Burt.

**Writing – review & editing:** Katie Willis, Austin Burt.

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
