## [Decision Letter · Decision Letter 0]

8 Feb 2021

Dear Authors,

We now have three detailed reviews of your manuscript. All of the reviewers have positive comments on your work. There is, however, a real need to revise the text to make the concepts and design more accessible to the general reader of PLOS Genetics. The reviewers provide some guidance. The manuscript is accepted with "minor revisions" because no further analyses are needed, but substantially improved accessibility of the material is a necessary condition for acceptance. This includes accessibility of the figures. One review had some concerns about whether this paper really broke new ground and another commented that it would have been useful (as well as traditional) in the introduction to mention other related work and concepts--the introduction could demonstrate the novelty of your ideas by presenting what came before them. One reviewer suggested further analyses to examine spatial structure, but also recognized that this could be beyond the scope of an initial paper--I agree. Finally, one reviewer requested that the model code be made available to others. This makes sense to me.

Your revision should be accompanied with a point by point response to each reviewer comm

Yours sincerely,

Fred Gould

Guest Editor

PLOS Genetics

Gregory Copenhaver

Editor-in-Chief

PLOS Genetics

Dear Authors,

We now have three detailed reviews of your manuscript. All of the reviewers have positive comments on your work. There is, however, a real need to revise the text to make the concepts and design more accessible to the general reader of PLOS Genetics. The reviewers provide some guidance. The manuscript is accepted with "minor revisions" because no further analyses are needed, but substantially improved accessibility of the material is a necessary condition for acceptance. This includes accessibility of the figures. One review had some concerns about whether this paper really broke new ground and another commented that it would have been useful (as well as traditional) in the introduction to mention other related work and concepts--the introduction could demonstrate the novelty of your ideas by presenting what came before them. One reviewer suggested further analyses to examine spatial structure, but also recognized that this could be beyond the scope of an initial paper--I agree. Finally, one reviewer requested that the model code be made available to others. This makes sense to me.

Your revision should be accompanied with a point by point response to each reviewer comment.

Reviewer's Responses to Questions

**Comments to the Authors:**

Reviewer #1: Synthetic gene drives have considerable potential for landscape-scale suppression of wild populations that carry disease agents (such as malaria) or cause significant environmental or agricultural damage (e.g. invasive pests). While there is considerable excitement about the prospect of gene drive deployment, there are also legitimate concerns about the potential impact of gene drive leakage into non-target populations. Recent studies have indicated that one possible strategy for target-population specific control is to exploit genetic differences in target and non-target populations – for example Sudweeks et al investigated if “private alleles” and be used for population suppression via single gene drive strategies. Such approaches are very sensitive to pre-existing resistance alleles – essentially the private allele needs to be completely fixed in the target population (but low levels in the non-target population can be tolerated). Here, Willis and Burt investigate if an alternative (more likely?) genetic architecture can be exploited – whereby the differentiated allele is present in the target population at high frequency (but not completely fixed) and not in the non-target population. Various double drives strategies are examined using a simple deterministic modelling approach. Some scenarios, even with relatively limited differentiation in the target population (e.g. 50% pre-resistance, Design 5), show quite remarkable suppression dynamics. Drive designs are realistic (based on existing studies at least in insects) and modelling parameters based on experimental data in An. Gambiae (noting that homing drives seem particularly efficient in Anopheles). It would be really interesting to see how the strategies perform with more realistic stochastic, individual-based (spatial) modelling but I accept their argument that this “first-pass” analysis.

The manuscript is clearly written, logically presented and is a significant body of work. I do not have any major concerns. The following points should be addressed.

1. Line 201-5 This is confusing – the text mentions “…killing heterozygous female (Fig 1 Designs 3 and 4)” – but Fig 1 indicates design 3 and 4 are haplosufficient (only design 5 is haploinsufficient). Further the Simoni et al [27] referenced gene drive is inserted into a haplosufficient gene.

2. Design 5 is a little unclear. The alpha locus drives autonomously in males (line 210) – this should be indicated in Fig. 1. Given there is only one Cas9 source (from the alpha locus) – then locus B should also be homing only in males. I cant see this explicitly mentioned and want to confirm this is what was modelled.

3. Very tight linkage between the alpha and beta loci could be confounded by the generation of large deletions (generated by end-joining after simultaneous DBSs) – this should be acknowledged in the Discussion.

4. Line 125-6 Some explanation should be provided regarding the strategy whereby functional resistance alleles will be avoided.

5. At lines 145-6, given an essential gene is not used in this context, it should be made clearer how the population rebound is occurring i.e. end-joining indels versus selection for the 1% non-target alleles that are present at the outset.

6. The significance of including the An. coluzzii data (Fig. 7) should be make clearer. It seems only An. gambiae is relevant. Also, why include the “uncertain” population in the legend?

7. Fig 7 shows a “proof-of-concept” analysis for the existence of “private allele” PAM sites in hypothetical target (island) versus mainland populations. Although the limited analysis is probably sufficient to make their point it would be interesting to know location of the PAMs, (i.e. the types of genes they reside in, particularly given that some strategies rely on an essential gene for the Beta locus), predicted off-target profile and on-target activity of their cognate protospacer sequences.

8. Line 358 A small caveat here that is hinted at but could be made more obvious is that the X-shredder could not be Cas9- based (if the homing constructs also use Cas9) – Cas9 is used for X-shredding in ref 43 and 44 but of course I-PpoI has been developed in An. Gambiae.

9. Given that homing rates in other species are generally lower and Anopheles, it would be interesting to see an analysis of a high performing drive strategy with cleavage/indel/homing rates from another well characterised insect – e.g. Drosophila melanogaster. Indeed, a sensitivity analysis for a high performing strategy would enhance the paper.

10. Line 78 The phrase “in the computer” is a touch awkward.

11. Line 197 Reference Sup Fig S4 here.

12. Line 431-2 Reference table S1 here.

13. What does “alpha-cargo” signify in Fig 6B?

Reviewer #2: This is an interesting paper in which methods (double drive) for local and transient (assuming extinction does not occur) population suppression, and modification, are described. Several other multi-component HEG-based systems have been previously described for modification. These are often somewhat baroque and have a number of requirements that may not be met in the real world in terms of target site conservation, etc. Other non-HEG-systems have also been proposed. This paper rises above this earlier work because it cleverly leverages the features of HEGs (in anopheles mosquitoes) that are known to work, in ways that make success likely. In particular, it leverages the fact of resistance allele formation in ways that are uniquely positive, rather than negative, as well as the previously reported ability to target the dsx locus without resistance allele formation, and to bias sex ratio using autosomal X shredders. In short, while this is a purely modeling paper, there is little doubt that multiple of the strategies described can be successfully implemented in Anopheles, and with some biological luck, elsewhere.

The simplest version of the idea is clear. An alpha element is designed to insert into and disrupt the female-specific functions of dsx, resulting ultimately in unfit homozygous females and fit males. This element is non-autonomous, containing either gRNAs or Cas9, but not both. A second element, beta, is inserted at a neutral locus that is enriched within the target population, and rare outside it. Beta is autonomous, and complements the missing alpha function, resulting in homing of alpha when both elements are present in the same individual. What makes this design (and related designs in which there is cross-complementation of various sorts) interesting is the fact that failure of the beta element to spread to all versions of its target site is guaranteed, and does not disrupt the intent. Beta's job is simply to increase in frequency enough to push alpha towards fixation, at which point population suppression ensues. Beta can either target a private allele, or in a more general version of the system, simply a neutral locus, with the inevitable appearance of resistant alleles ultimately limiting the spread of it and alpha.

Thus, the key here is that with the designs proposed the authors are able to take advantage of what we (the field) already know how to do really, really, well, which is to build a HEG that fails due to resistance allele formation or the presence of pre-existing polymorphisms that are resistant alleles. The designs are plausible, because they utilize the known: consistent targeting of dsx, inconsistent targeting of pretty much any other site, either due to pre-existing polymorphisms, or new resistant allele formation, and X shredding-based sex ratio bias.

My one specific request is that the authors provide the code on github for their work, along with an explanation of how to use it. Referencing back to a paper from 2008, which also does not provide code, does not provide sufficient guidance to understand and explore the many variables, loci and alleles worked with in the current manuscript. It is necessary that others be able to not only read the text, but also work with tools that allowed its creation. In the absence of these methods the paper is hard to explore, and it is not possible to test other scenarios. The details of how the model was implemented are undoubtedly very interesting. It clearly involved a lot of work (which remains invisible in the background otherwise), as it involves many alleles at multiple loci, shredding, homing and recombination. One hopes that general models such as this that can handle a lot contribute to the field eventually adopting a coherent modeling platform rather than a series of lab-specific one-offs.

The manuscript is very dense and almost always asks the reader to already be intimately familiar with the behavior of related systems. The text describing the different designs and outcomes reads a bit like a math textbook. There is no walk through for the uninitiated, and sections typically end with a statement of outcome, in much the way a math text book would end a proof with the phrase "from inspection it can now be inferred that...", which then leads to a figure and a turn to a new chapter.

Or to put it another way, in reading and re-reading the text and very dense figures (which rarely or never guide the reader through the data), I was reminded of the first line from Ludwig Wittgenstein's Tractatus, Logico Philosophicus: "This book will perhaps only be understood by those who have themselves already thought the thoughts which are expressed in it—or similar thoughts."

I would like to strongly suggest that the authors make more of an effort to walk readers through the designs and how they work. The figures are very dense and key features of the panels are never discussed. The figures are simply presented, along with very minimal text, and the authors then move on to the next section, with the implicit understanding that the reader will somehow muddle through.

Other suggestions are below. Many of these also relate in one way or another to expanding discussion to make the work more accessible to non-specialists. The ideas presented are really beautiful and plausible, and I just think that they can be made more accessible in ways that will allow others to better intuit the forces at work and how the ideas presented could be extended further. The manuscript is not overly long, and is online only in any case. If the authors feel strongly for some (compelling) reason that they want to keep the text tight and telegraphic they could also consider an extended supplementary discussion of the various systems. A nice example of this is the remarkably complete supplemental notes file provided in another article from this group "How driving endonuclease genes can be used to combat pests and disease vectors"

In the introduction the authors might consider introducing other double drives from the literature. This would allow the authors then to begin drawing distinctions between prior goals and their own, and how their constructs etc therefore differ. See related comments on discussion.

Its also a bit hard, in figures like 2 and other similar figures, to keep track of what all the lines mean. It requires going back and forth between the figure legend and Figure 1. It would be nice to introduce these a bit more visually, perhaps in a box at the top of figure 2. And again, none of this is discussed in the text.

It might be nice if the authors would note at some point that the idea of using linkage to increase drive strength has been noted and explored to some extent in other systems.

In figure 3 it would be good to walk the reader a little bit through the significance of targeting a haploisufficient locus that manifests phenotypes in particular sexes/genotypes. Again, this is one of those points in the text where the reader really needs to have a deep familiarity with the prior literature in order to understand how and when this manifests itself, and how this contributes to suppression while still allowing strong drive.

In the legend for figure 3 it is not immediately clear what is going on with the single locus drive and why it works so much less well. Since a single locus drive is presumably inserted into the dsx locus, which thus far does not accumulate resistance alleles, what causes the failure? The legend makes no mention of these or other potentially important variables. Is there a resistance allele in there somewhere that is not mentioned? If so, why is it present in single locus dsx, but not the alpha locus of the two locus versions? Legends should include all the relevant variables at work.

In line 200 it would be nice to explain in a bit more detail why the x shredder kills heterozygous females. When is it expressed and how does this still allow rapid and strong drive

In lines 211 onwards this is the first place where the authors make an attempt to walk the reader a bit through how the system, a particularly complicated one, does its job.

Line 239. This is an odd reference for the idea that hegs have reduced replication fidelity. While the referenced construct does have reduced stability due to the presence of repeats, this is not the same thing as the reduced fidelity that may be associated with homing per se. Is it appropriate to also reference some of the evidence suggesting HR repair of a ds break may lead to higher frequency of mutation, or do you just mean to focus on the repetitive components, gRNAs, if multiple are used? Though if dsx is being targeted with a single gRNA then it is not clear there is a repetitive element.

For Fig 6 I don't quite understand the lines in panel B. In particular why is the frequency of alpha different from that of alpha-cargo. The alpha construct contains the cargo, so shouldn't they be the same? This figure is also one where a bit more captioning in the figure itself would help the reader. Finally, If alpha-cargo is the entity of interest, wouldn't it make more sense to show its genotype frequency rather than allele frequency, or at least both? The allele frequency appears to saturate at about 65-70%, but it is unclear what this means in terms of carrier frequency.

In Fig. 6C the non-target population seems to perhaps rise to a significant number, maybe 30%. It is hard to know what the number is since its a log scale. It would be useful here and perhaps a few other places to provide actual numbers so the reader can evaluate this, since non-target population effects are the topic that tends to get folks excited.

The data presented on private allele frequencies is intriguing, and the authors are appropriately cautious about its interpretation given the limited data available.

In lines 358-353 in the discussion, the authors bring up, for the first time, other two locus systems. Some, such as myself, would have placed this and more in a manuscript introduction that put the current work on two locus systems into a historical context that discussed what had come before, including other two locus homing systems. The authors have chosen not to do that, or provide any more than a quick reference nod to this and other literature in their discussion. It is an interesting strategy, as it keeps the readers attention squarely focused on the authors' thoughts and accomplishments, while minimizing any mental clutter from the literature that might interfere with this task. That said, while the prior literature is not discussed, its existence is noted, and thus represents a stylistic choice that is acceptable.

In line 363 the authors discuss the idea of adding extra gRNAs to the alpha construct. They say this will increase the load, but they dont explain how this will work. Can they do this, briefly? Spitting off LOF mutations (or HEGs?) at independently segregating loci does this because......? Again, the diligent reader can at some point come up with a plausible scenario, but the authors could guide them to this point directly in a few lines. The two references provided may not be the most relevant. One is an old paper that talks about several uncloned maternal effect mutants and their effects on embryo development, and the other discusses roles of fruitless in males, not females. Is there something more relevant based on cloned genes with known behavior that could be cited? In any case, the key point is to walk the reader through how these alleles bring about increased load while still allowing drive.

On line 144 the authors refer to dominant fitness costs. Can they clarify that this does not mean (I assume) completely dominant, as in heterozygote fitness cost = homozygote fitness cost. If it does mean completely dominant can they explain the basis for this, as I would normally imaging fitness costs to be additive in some sense. Are they meaning to imply that off target cleavage effects are 100% regardless of the dosage? That might explain it.

Reviewer #3: Willis and Burt present a simulation study of gene drive approaches that could potentially be localized to specific target populations. They focus on a combination of two basic concepts: two-locus drives (aka “double drives”) and the targeting of “private” alleles. The key idea is that there is a division of labor between the constructs at the two loci, with the first responsible for the desired impact (population suppression or replacement) and the second for the localization, achieved through the targeting of a private allele. The first construct can only drive when the second is present. In the target population, both constructs are therefore expected to drive, whereas they would resemble a split drive in non-target populations. The paper conducts computer simulations of a number of different such strategies.

This is a well-written paper on an interesting topic. It is definitely more plausible that a real-world use case of a gene drive might exist for a type of drive that can be localized, as compared to standard homing drives that have the potential to spread around the world. The inclusion of an analysis of actual polymorphism data in a mosquito species to see how common the required “private” alleles are observed is quite interesting as well. I think we generally expect these sorts of sequence variations to be present between differentiated populations, and I wouldn’t say that it is critical that a modeling paper include this kind of analysis, but it’s interesting to see.

My main comment is concerned with the extent to which this study provides an advancement over previous studies that have already provided a proof-of-principle of double drives achieving localization by targeting private alleles (notably, the recent study by Sudweeks et al). The authors of the current study are very clear that their study is still intended to serve as a proof-principle, rather than an assessment of an actual potential application, which would certainly require more detailed and realistic modeling to be meaningful (although they do study specific populations of An. gambiae in their analysis of polymorphism data). As such, the current study appears to re-tread some conceptual ground that has already been explored in previous studies. Obviously, there are some differences between the specific drive designs studied in their paper and Sudweeks et al. The latter study also focused on sequences that are fixed in the target population, but absent or at low frequency in the non-target population (the opposite of the current study). Whether these differences constitute enough advancement for the journal is a subjective question I want to leave to the editor to decide.

Another (admittedly very general) concern is how confident we can be that even basic, qualitative conclusions from this study will ultimately hold in a real-world population, given the rather simplistic assumptions made by the modelling. In particular, it is unclear whether results from a panmictic population model still apply for realistic populations that are inherently spatially structured, for example, because they occupy a continuous landscape with limited dispersal. This can affect dynamics not only quantitatively, but can give rise to new qualitative phenomena that fundamentally change the outcome of a drive (e.g. resulting in a suppression drive failing to suppress a population). At the core of such new phenomena often lies the heterogeneity that can arise in spatial populations, which doesn’t exist in panmictic models. I would expect that two-locus drive strategies would be particularly prone to such issues, as compared to single locus strategies. This is due to the assumption made by panmictic models that the frequency of allele combinations at the two loci is simply the product of their individual population frequencies. In a spatial population, however, the two alleles could spread at considerably different rates, and thus be present very heterogeneously. The panmictic model would then rely on fundamentally wrong predictions of how frequently such alleles would actually “meet” in an individual based on their overall population frequencies, and this could certainly affect the dynamics profoundly.

I admit that a comprehensive analysis of such questions would be open-ended and probably beyond the scope of the current paper. The authors do already discuss several other limitations of their model in the discussion section. Maybe a mention of the potential impact of spatial heterogeneity and limited dispersal could be added as well.

Minor comments:

The authors might consider reworking and extending the methods section. It is very short and seems a bit rushed compared to the rest of the paper. Some parameters are specified in this section, while not really enough detail is provided about what those parameters do. E.g. on line 428-430:

“juvenile survival is density dependent according to the Beverton-Holt model, which has two parameters, but since we report results in terms of relative population sizes, only one matters, the intrinsic rate of increase (Rm).”

Somehow this sentence simultaneously tells me what specific implementation the model uses, while telling me very little about that implementation, and tells me that there are two parameters, and then tells me that there is only one parameter. It also seems a little awkward that the PAM material and the modeling material are not in separate paragraphs. I am aware that these authors focused more attention on the supplemental methods section, but I feel that if there is going to be a short methods section in the main body, it should focus on giving a rough sketch of how the model works without going into unnecessary specifics, but ideally at least introducing all aspects of the model.

2. I find the statement that their model is individual-based a bit misleading. The model used portions of the total starting number of individuals who are in a given state, and as the population declines, the simulation isn’t tracking an actual simulated population size, but rather, a size relative to the equilibrium size. I’m sure this allows for simulations to be performed with great computational efficiency, and no doubt this type of model has its place in gene drive research. Still it is not truly an individual-based model, resulting in somewhat awkward situations such as that actual population elimination is never possible (as frequencies can become arbitrarily small but never truly reach zero).

3. The caption of Figure 2 seems to be missing some details (e.g. it doesn’t state that the dashed lines indicate resistance). The figure caption could probably be simplified a lot if a legend were included with the figure. In general, I feel that several of the figures might benefit from having legends, especially because the same style of line means different things from figure to figure (e.g. in Figure 2, the dashed lines indicate resistance within the population, in Figure 3 the dashed lines indicate that construct β is inserted in a haplo-insufficient target). Including a legend would probably make the figures a fair bit easier to digest.

**Have all data underlying the figures and results presented in the manuscript been provided?**

Reviewer #1: Yes

Reviewer #2: None

Reviewer #3: Yes

PLOS authors have the option to publish the peer review history of their article (what does this mean?). If published, this will include your full peer review and any attached files.

Reviewer #1: No

Reviewer #2: **Yes: **Bruce A Hay

Reviewer #3: No

---

## [Editor Report · Decision Letter 1]

7 Mar 2021

Dear Dr Willis,

We are pleased to inform you that your manuscript entitled "Double drives and private alleles for localised population genetic control" has been editorially accepted for publication in PLOS Genetics. Congratulations!

Yours sincerely,

Fred Gould

Guest Editor

PLOS Genetics

Gregory P. Copenhaver

Editor-in-Chief

PLOS Genetics

Comments from the reviewers (if applicable):

The authors have addressed all of the reviewers comments either by making changes to the manuscript or by explaining why changes are not needed or not appropriate. As such, I find the manuscript ready for publication.

**Data Deposition**

http://datadryad.org/submit?journalID=pgenetics&manu=PGENETICS-D-20-01914R1

**Press Queries**

---

## [Editor Report · Acceptance letter]

18 Mar 2021

PGENETICS-D-20-01914R1 

Double drives and private alleles for localised population genetic control 

Dear Dr Willis, 

We are pleased to inform you that your manuscript entitled "Double drives and private alleles for localised population genetic control" has been formally accepted for publication in PLOS Genetics! Your manuscript is now with our production department and you will be notified of the publication date in due course.

With kind regards,

Katalin Szabo

PLOS Genetics

On behalf of:
